# The Initial Exploration Problem in Knowledge Graph Exploration[*]

Claire McNamara[1,*,†], Lucy Hederman[1] and Declan O'Sullivan[1]

[1]*School of Computer Science and Statistics, Trinity College Dublin, Dublin, Ireland*

## Abstract

Knowledge Graphs (KGs) enable the integration and representation of complex information across domains, but their semantic richness and structural complexity create substantial barriers for lay users without expertise in semantic web technologies. When encountering an unfamiliar KG, such users face a distinct orientation challenge: they do not know what questions are possible, how the knowledge is structured, or how to begin exploration. This paper identifies and theorises this phenomenon as the Initial Exploration Problem (IEP). Drawing on theories from information behaviour and human-computer interaction, including ASK, exploratory search, information foraging, and cognitive load theory, we develop a conceptual framing of the IEP characterised by three interdependent barriers: scope uncertainty, ontology opacity, and query incapacity. We argue that these barriers converge at the moment of first contact, distinguishing the IEP from related concepts that presuppose an existing starting point or information goal. Analysing KG exploration interfaces at the level of interaction primitives, we suggest that many systems rely on epistemic assumptions that do not hold at first contact. This reveals a structural gap in the design space: the absence of interaction primitives for scope revelation, mechanisms that communicate what a KG contains without requiring users to formulate queries or interpret ontological structures. In articulating the IEP, this paper provides a theoretical lens for evaluating KG interfaces and for designing entry-point scaffolding that supports initial exploration.

## Keywords

Knowledge Graphs, Human-Computer Interaction, Exploratory Search, Knowledge Graph Exploration, Knowledge Graph Usability, Initial Exploration Problem

## 1. Introduction

A Knowledge Graph (KG), as defined by Hogan et al. [1], is a structured graph that models real-world knowledge by representing entities as nodes and the relations between them as edges. A *node-edge-node*, otherwise known as *subject-property-object*, is called a triple and it is possible to store this information on computers through the use of the Resource Description Framework (RDF) [2]. Enabling the linkage of data in this manner allows for interoperability beyond traditional information silos into a semantic web of knowledge. Examples span domains. In medicine, the bridging of seven European data registries as carried out by the FAIRVASC project [3] enables clinical researchers to review aggregated patient information across countries giving insight into rare kidney disease. In the encyclopedic domain, DBpedia [4], a large-scale KG (as of the 2016 release, containing 9.5 billion triples [5]) extracts and structures information from [6] information boxes. In the digital humanities domain [7], the Virtual Record Treasury of Ireland (VRTI) KG [8] storing and linking historical people, places, and more recently, events (entities), provides another example. Created by historians and computer scientists as part of the VRTI project [9] it supports the reconstruction of records that were lost in the destruction of the Public Record Office of Ireland during the Irish Civil War in 1922. Such KGs typically use a variety of ontologies to model the information contained within them. For example, the VRTI KG makes use of, and at times extends, CIDOC CRM [10], a widely adopted cultural heritage ontology standard, GeoSPARQL [11], a standard for representing and querying geospatial data, and its own custom VRTI ontology [12]. The resulting KGs are richly interconnected and semantically expressive, but such richness also brings complexity often requiring a level of technical expertise in order to query over them [13]. We therefore

*Second International Workshop on Users and Knowledge Graphs (UKG) @ Semantics 2026*

[*]Corresponding author.

✉ mcnamacl@tcd.ie (C. McNamara); hederman@tcd.ie (L. Hederman); declan.osullivan@tcd.ie (D. O'Sullivan)

🆔 0000-0002-8263-8694 (C. McNamara); 0000-0001-6073-4063 (L. Hederman); 0000-0003-1090-3548 (D. O'Sullivan)

define a complex KG as one whose underlying ontology introduces semantic and structural complexity that makes the graph difficult to interpret and explore without prior ontological or technical expertise.

As such, there exist significant barriers to entry between the KG and lay users, and unless the relevant people e.g. clinicians, literary scholars, or any lay user [14] (an ordinary web-user, i.e. an individual who does not possess technical knowledge of RDF-related technologies) can explore them, the benefits of representing knowledge in this way may not readily surface. Lay users therefore face a clear orientation barrier when first engaging with a KG: they may not know what questions are possible, how to map those questions to the structure outlined in the ontology, or how to express them in a formal query language. This first-contact problem can prevent users from taking their initial steps in exploration and, unless mitigated, risks leaving large-scale KGs inaccessible to the broader communities they aim to serve.

This paper is a conceptual and positioning contribution. It synthesises existing strands of work to conceptualise a previously under-specified barrier in KG interaction and proposes a design-oriented framing intended to guide future empirical investigation and system development. We first present a conceptual gap in Human-Computer Interaction (HCI), Semantic Web, and Knowledge Graph (KG) usability literature, where the challenges experienced at the onset of exploration of a large, complex KG have been addressed in a fragmented manner. We then introduce a conceptual framing for this "first-contact" problem, which we refer to as the Initial Exploration Problem (IEP). We then examine existing KG exploration interface design primitives through the lens of the IEP framing, highlighting the gap under-addressed through traditional means. In defining IEP and characterising it through a conceptual framing, we argue that both clearer evaluation criteria and the development of targeted scaffolding techniques are enabled, not only in support of our ongoing research, but in support of researchers and practitioners in the community.

This paper makes four contributions. First, it defines the Initial Exploration Problem (IEP) as a temporally bounded pre-goal state that arises at first contact with unfamiliar KGs. Second, it characterises the IEP through three interrelated barriers: scope uncertainty, ontology opacity, and query incapacity. Third, it identifies scope revelation as a missing interaction primitive required to address this state. Fourth, it outlines design implications and an evaluation lens to support future empirical validation.

Throughout this paper, the term *IEP* is used in two closely related but analytically distinct ways. Firstly, it names a recurring empirical problem observed when lay users encounter an unfamiliar, complex KG and struggle to identify how or where to begin exploration. Secondly, we develop a conceptual framing of the IEP that explains the problem by characterising the conditions under which it arises. Where necessary, we distinguish between the IEP as a named problem and the conceptual framing of the IEP composed of specific, interdependent characteristics.

## 2. The Theoretical Underpinnings of the Initial Exploration Problem (IEP)

Whilst the barriers arising from the IEP are closely linked to the difficulties associated with lay users interacting with KGs, such difficulties can be better understood through looking at existing HCI and information behaviour concepts and theories. This section examines these concepts and theories in an effort to show why no one existing term fully captures the IEP and sheds light on how the IEP has previously been explored in fragments across the literature.

### 2.1. Foundations in Information Behaviour: Uncertainty and ASK

At the onset of user interaction, exploration is fundamentally different from traditional information retrieval (IR) [15, 16]. Traditional IR typically assumes that the user has an information need that they are seeking to fill through asking a question over information [17]. Exploration in contrast does not assume such a need; the user instead is making their way through the data in what could be described, at least at first, as an open-ended fashion driven by curiosity as opposed to explicit questions. Through

their exploration, questions may occur to the user at which point traditional IR comes into effect. More specifically, once on a path, the user can make decisions on where to go next, traversing the web of knowledge provided to them. The IEP specifically refers to the empirical difficulty observed at the onset of exploration, where curiosity is present but a goal is absent. The IEP framing characterises the difficulty of identifying a starting point for traversal, and the ways in which this difficulty is exacerbated by the format in which knowledge is represented (i.e., KGs).

The state of uncertainty of where to begin is analogous to Belkin's Anomalous State of Knowledge (ASK) [18] where he identifies users looking at IR from a place of "perceived wrongness". Essentially, that the user's current internal picture of their knowledge is flawed and they are looking to correct it, therefore mapping the problem to IR. The difference captured by the IEP framing is that the user's sense of uncertainty does not come from a gap or flaw in their internal knowledge that they are looking to fix but instead, the uncertainty comes from a combination of challenges, namely the characteristics that make up the IEP: the user does not know the scope of the KG, they do not know how the data is structured, and they do not have the relevant expertise in using SPARQL in order to orient themselves. ASK presupposes a knowledge gap; the IEP precedes the ability to articulate such a gap. Where ASK assumes a misalignment between what a user knows and what they need to know, the IEP describes a prior state in which neither the nature of the gap nor the means of resolving it are yet apparent.

## 2.2. Frameworks for Process: Exploratory Search and Sensemaking

Exploratory search is an umbrella term under which KG exploration sits and is encountered here as exploration preceding information need. More specifically, KG exploration can be understood as a specific instantiation of exploratory search, characterised by explicit semantic structure and graph-based interaction. Marchionini [19] captures exploratory search as a combination of learning and investigating but does not isolate a user's first contact with data; the assumption is that the user can begin "somewhere" and tools then should support their exploration journey. The problem with this is that when a lay user has no underlying mental model of the structure of the data or what they can ask, such a starting point is not obvious. A similar limitation arises with Russell et al.'s sensemaking paradigm [20]; in this instance, there is the presence of a goal or "task", something which is at odds with exploration driven by curiosity. Similarly, while White et al. [21] acknowledge exploratory search can be driven by curiosity, in the absence of a goal, they do not investigate or propose a starting point from which curiosity-driven exploration can begin.

## 2.3. Cognitive Foundations: Load, Foraging, and Scaffolding

Information Foraging Theory [22] provides a complementary lens, modelling information seekers as predators following "information scent". The IEP represents a complete absence of scent; without prior knowledge of the KG's scope or structure, users cannot evaluate the utility of potential paths, leaving them "stranded" at the entry point with no trajectory to follow. This absence of information scent has direct implications for cognitive load. At first contact, users must simultaneously interpret unfamiliar terminology, infer structural conventions, and evaluate potential entry points, resulting in high intrinsic and extraneous cognitive load [23]. Unlike later stages of exploration, this load cannot be mitigated through strategy or experience.

## 2.4. Where do we Start? Orientation, Onboarding, and Serendipitous Discovery

When considering the issue of "where to begin" as it relates to human interaction with digitised data, the HCI concepts of orientation and onboarding are surfaced, referring to situations where users need initial guidance to build situational awareness in information systems. Onboarding in User Interface (UI) design emphasises progressive disclosure, greeting users with simple tours or anchors to reduce abandonment. In KG contexts, this could involve default starting entities or visual maps. However, onboarding is a method of solution to the IEP, it does not articulate the problem. In short, onboarding describes how systems introduce users to interfaces; the IEP describes the epistemic condition that

makes such introduction necessary in the first place. Serendipitous discovery complements onboarding: while IEP blocks entry, tools fostering unexpected finds (e.g., following paths [24]) can spark exploration once initiated [25].

## 2.5. Synthesis

These underpinnings can be synthesised into a cohesive, clearly scoped account of the problem by isolating a distinct, first-contact condition in KG interaction: it precedes ASK by embodying pre-gap uncertainty, extends exploratory search by isolating the entry barrier, and leverages cognitive theories for solutions like scaffolded entry-points. Unlike sensemaking's task-focus, the IEP framing highlights curiosity-driven starts in semantic spaces. Table 1 provides a comparison of where the IEP framing sits in relation to cognitive and usability literature. Although the concepts listed in the table originate from different theoretical traditions (ranging from cognitive psychology to interaction design), the purpose of listing the concepts is not to equate them analytically, but to position them temporally within the user interaction lifecycle. Specifically, the table examines whether each concept presupposes that a user has already succeeded in initiating meaningful interaction with a KG.

**Table 1**
A comparison of the Initial Exploration Problem (IEP) framing with related concepts in information seeking and interaction.

| Concept | Primary focus | Assumes a starting point? | Assumes a goal or question? | Applies at first contact? |
|---|---|---|---|---|
| Exploratory search | Learning, investigation, discovery | Yes | Not necessarily | No |
| Sensemaking | Constructing understanding from information | Yes | Yes (explicit or implicit) | No |
| Information foraging | Navigational and information-seeking strategies | Yes | Implicitly | No |
| Berrypicking [26] | Iterative query refinement | Yes | Yes (evolving) | No |
| ASK (Anomalous State of Knowledge) | Knowledge gap triggering search | Yes | Yes | No |
| KG usability issues | Interaction and system-level barriers | Yes | Often | Partially |
| Serendipitous Discovery | Unexpected and valuable discoveries during exploration | Yes | No | Sometimes |
| Information overload | Cognitive effect of excessive information | Yes | Not required | Sometimes |
| **Initial Exploration Problem (IEP)** | **Entry into exploration of an unfamiliar KG** | **No** | **No** | **Yes** |

Building on these theoretical fragments, the next section demonstrates how the Initial Exploration Problem (IEP) manifests as a unified, observable phenomenon in KG contexts.

# 3. The Manifestation of the Initial Exploration Problem (IEP) in KGs

This section demonstrates that the problem referred to as the IEP is a real, observable phenomenon in the KG exploration domain by analysing the causes identified through the IEP framing. In the first three subsections, it illustrates how the IEP manifests in practice through recurring epistemic and interactional barriers observed in KG exploration research. Informed by these barriers, the next two subsections proceed to present our proposed framing of Initial Exploration Problem (IEP) in terms of three inter-dependent barriers with respect to interaction of users with KGs.

## 3.1. Complex Ontologies and Technicality

At first contact, complex ontologies do not merely impede query formulation; they prevent lay users from forming even a coarse understanding of what the KG contains, and therefore from identifying a plausible point of entry. Prior work such as [27] recognises the complications incurred in beginning to explore an unfamiliar KG by a lay user: the user's mental model of the underlying structure of the knowledge does not match the true complex structure of the ontology. This leads to overwhelm and an inability to query effectively: "users do not know what they do not know" [18]. Moreover, there is an assumption that users understand sufficiently the domain in which they are beginning to explore to, if provided with facets or entities, be able to choose ones that will provide them with meaningful exploration starting points [28]. An assumption that does not hold in practice and constitutes as a direct manifestation of **scope uncertainty**. Furthermore, without an understanding of the ontology or conceptual schema underlying the KG (**ontology opacity**), lay users are unable to anticipate how information is organised or what forms of questions are meaningful. This disconnect between the user's mental model and the system's semantic structure has been repeatedly identified as a barrier to effective KG interaction [29] and as such, **ontology opacity** manifests not only as a difficulty executing actions, but as a difficulty anticipating what kinds of questions and relationships are meaningful within the KG.

It has long been established within KG usability literature that there is a significant barrier between lay users and the technical query language SPARQL [30]. This has led to numerous solutions aiming to hide the complexities of SPARQL behind more intuitive interfaces [31, 32]. However, such solutions do not look at the moment of first contact; they become effective only after the user knows in some capacity what it is that they are looking for. At first contact, **scope uncertainty** precedes **query incapacity**: users struggle not only to express queries, but to determine what kinds of questions are possible to be asked in the first place.

## 3.2. Scale and Interconnectedness

Scale and interconnectedness do not introduce new barriers, but intensify the epistemic opacity faced at first contact, making it harder for users to infer what constitutes a promising direction of exploration. For instance, large KGs like DBpedia [5], with billions of triples, exemplify how the problem is exacerbated by presenting a hyper-connected space where every entity links to dozens or hundreds of others, leading to confusion and high cognitive load [14]. This is further discussed by Al-Tawil et al. [27] with the authors noting that "users can face an overwhelming amount of exploration options and may not be able to identify which exploration paths are most useful" which "can lead to confusion, cognitive load, frustration, and feeling of being lost". This results in a profound difficulty of understanding the scope (**scope uncertainty**) of the KG by lay users.

Furthermore, while lay users may enter with vague preconceptions or hypotheses about potential content (e.g., assuming a historical KG contains information on well-known historical events), these assumptions can mislead: what they believe might exist may not (leading to fruitless searches), or if it does, it may represent a peripheral or isolated subset of the KG, resulting in exploratory "dead ends" obscuring the true breadth and interconnectedness of the KG. While the IEP is most visible in curiosity-driven exploration, it also applies when lay users possess only vague or underspecified goals, insofar as they lack the knowledge required to identify a meaningful entry point into the KG.

### 3.3. The IEP in the Digital Humanities (DH) Domain

An example of how the IEP can be observed exists within the Digital Humanities (DH) domain. This is due to the often complex ontologies used (e.g. CIDOC CRM [10]), the scale of the KGs in question (e.g. the VRTI KG at 2.9 million triples [8]), and the persistent difficulty of the query language SPARQL [33] for lay users. Studying the IEP in the DH is therefore an ideal use case: the KGs themselves are rich and expressive, but the IEP may hinder their exploration by lay users. For lay users, these characteristics compound **scope uncertainty** and **ontology opacity**: not only is the domain unfamiliar, but the intellectual organisation of the knowledge itself is opaque, making it difficult to infer what kinds of historical questions the KG can meaningfully support.

### 3.4. The Characteristics of the IEP

Within the proposed framing, the IEP is characterised as the co-occurrence of three interdependent barriers that arise simultaneously at first contact and can be distilled as:

**Scope Uncertainty:** Scope uncertainty represents the core problem of "where to begin". Where do you start if you have no knowledge of what is contained within the KG? This is exacerbated in large-scale KGs where users cannot gauge the breadth of topics, leading to cognitive overload [14] and is therefore epistemic rather than technical: it concerns the user's inability to reason about the knowledge space itself.

**Ontology Opacity:** The structured nature of a KG promotes interoperability which is one of the reasons as to why they are so valuable for knowledge representation. However, this structure can often be difficult for lay users to interpret and map mentally [34, 29], particularly in the absence of exemplar queries or conceptual anchors.

**Query Incapacity:** SPARQL as a query language has long been acknowledged as a significant barrier to lay users utilising KGs [35, 36, 34, 30].

### 3.5. The IEP Framing as a Unified Concept

Across these manifestations, scope uncertainty emerges as the dominant first-contact breakdown: ontology opacity and query incapacity become problematic precisely because users lack an initial understanding of what the KG contains and how exploration might productively begin. Even so, the characteristics within the IEP framing do not exist in isolation, all are present at the onset of exploration of an unfamiliar, large, complex KG by a lay user. Addressing one does not alleviate the other two; for example, providing users with a means of asking NLQs over a KG does not address the problem of what to ask (**Scope Uncertainty**) or, how to structure the question (**Ontology Opacity**). Similarly, in knowing what one wants to ask, one does not necessarily have the ability to structure the question in a manner that aligns with the underlying structure of the KG (**Ontology Opacity**) or have the necessary technical skill to create SPARQL queries (**Query Incapacity**). It is therefore important to consider the problem holistically when designing solutions that aim to address it.

More concretely, it is possible to understand the problem as the following:

*The Initial Exploration Problem (IEP) is the temporally bounded, pre-goal state experienced by a lay user at first contact with an unfamiliar, complex KG, characterised by the simultaneous presence of scope uncertainty, ontology opacity, and query incapacity, which together prevent the user from identifying a meaningful starting point for exploration.*

Existing work provides essential foundations but does not explicitly address the pre-goal convergence of scope uncertainty, ontology opacity, and query incapacity at first contact. Exploratory search research [19, 21] addresses evolving goals but typically presumes accessible content structures. KG usability studies document schema comprehension barriers but often assume predefined tasks [29, 34]. Generous interface approaches foreground orientation but do not formalise the interactional state that precedes meaningful engagement [37, 24]. The IEP integrates these strands into a unified framing focused on initial engagement conditions.

# 4. The IEP as a Constraint on KG Interface Design

The IEP is temporally-bounded: it presents itself at the very start of KG exploration. It is because the problem is temporally-bounded that addressing it requires targeted solutions distinct from those addressing ongoing exploration. This paper argues that addressing the first contact challenge in KG exploration in isolation is necessary as it is one that, unless considered as a problem in its own right, may lead to solutions developed that do not adequately address the barriers encountered. Crucially, this difficulty is not limited to identifying an interface action, but reflects an inability to determine what constitutes a meaningful or valid starting point within the knowledge space itself. In particular, it suggests that many existing interfaces fail to address the IEP not because of shortcomings in implementation, but because of the fundamental design assumptions embedded in their interaction models.

Rather than examining individual systems as isolated artefacts, this section analyses KG exploration interfaces at the level of interaction primitives and epistemic dependencies. That is, it asks what kinds of user knowledge an interface assumes must already be in place in order for interaction to begin. This shift in unit of analysis allows the IEP framing to function as a design constraint on the space of possible interface solutions, rather than as an evaluative checklist applied post hoc.

## 4.1. Solution Development Pre-Problem Formulation

In the broader traditions of Human-Computer Interaction (HCI) and Semantic Web research, it is not uncommon for practical systems to be developed in response to recurring user difficulties before those difficulties are formally conceptualised as distinct research problems. This phenomenon, where design precedes theoretical articulation, has been noted in several areas of computing and information science. For instance, the emergence of recommender systems in the late 1990s and early 2000s [38] responded to the growing practical problem of information overload on the web. Only in subsequent years was the challenge rigorously theorised within information behaviour research as a cognitive and affective phenomenon [39]. Similarly, in NLQ interfaces to databases and KGs, early tools such as START [40] and more recent QA systems like PowerAqua [41], as well as later QA evaluation campaigns such as QALD [42], addressed usability barriers long before formal taxonomies of query intent [43], exploration vs. lookup [19, 21, 17], or onboarding challenges [14] in KG interaction were widely discussed in academic literature.

Naming and characterising the IEP is further motivated by the development of systems such as OnSET [44] and Knowledge Anchors [27]. Such systems can be understood as independently developed responses to a shared but unnamed challenge: enabling users to initiate meaningful exploration of complex KGs. While these approaches differ in modality (visual, conceptual, and cognitive), they each introduce scaffolding intended to overcome the same initial barrier faced by lay users. Upon closer inspection, however, a gap in conceptualisation becomes evident. Al-Tawil et al.'s [27] approach of Knowledge Anchors frames KG interaction as a learning process leaning on subsumption theory [45], the idea that people learn better by associating new knowledge with the knowledge they already possess. This theory however assumes that a user can select a meaningful starting point with which to build from. The IEP framing makes this assumption explicit and problematises it, thereby providing the missing theoretical foundation for such practical interventions.

## 4.2. Interaction Primitives and Epistemic Preconditions

It is possible, drawing on the work of [46, 47], and [48], to decompose any KG exploration interface into a small number of interaction primitives such as searching, filtering, selecting, browsing, or querying. While these primitives differ in modality (textual, visual, graphical), they share a common property: each presupposes certain epistemic conditions on the part of the user. For example, issuing a search query presupposes that the user has and can articulate a meaningful information need, while selecting a class or facet presupposes some awareness of the underlying schema or domain structure and what those classes or facets mean.

Within the IEP framing, these presuppositions become problematic. At first contact, a defining characteristic of the IEP is precisely the absence of such conditions: the user does not yet know what the KG contains (scope uncertainty), how it is structured (ontology opacity), or how to express questions in a formal language (query incapacity). As a result, interaction primitives that rely on any of these forms of prior knowledge are structurally misaligned with the moment the IEP arises.

Table 2 summarises common interaction primitives used in KG exploration interfaces and the epistemic assumptions they impose.

**Table 2**
Interaction primitives in KG exploration interfaces and their epistemic preconditions.

| Interaction primitive | Primary function | Epistemic precondition |
| --- | --- | --- |
| Keyword or entity search | Locate relevant entities or facts | User has and can articulate a meaningful query or term |
| Facet selection | Narrow result space | User understands facet semantics and relevance |
| Schema or class browsing | Reveal ontology structure | User can interpret abstract ontological concepts |
| Graph navigation | Traverse linked entities | User has an initial conceptual anchor |
| Visual query construction | Build structured queries | User understands relationships and constraints |
| NL question answering | Retrieve answers via natural language | User can articulate an information need in natural language |

Viewed through this lens, it becomes apparent that most existing KG exploration interfaces are designed around primitives that presuppose the successful resolution of at least one of the IEP characteristics. Consequently, they are effective only after the IEP has been overcome, either through prior knowledge or external guidance.

### 4.3. Paradigms of KG Exploration as Compositions of Primitives

Existing KG exploration systems can be understood as compositions of the interaction primitives outlined above. Faceted browsers (e.g., KG Explorer [49]) combine search and filtering; visual exploration tools (e.g., Ontodia [50], KGViz [51], and PG-Explorer [52]) combine schema exposure and graph navigation; natural language interfaces (e.g., SPARKLIS [32]) combine question formulation with query translation. While these paradigms differ in surface interaction style, they are unified by a shared assumption: that the user is already capable of selecting a meaningful starting point.

From the perspective of the IEP framing, this assumption is precisely what fails at first contact. Interfaces that prioritise navigation, refinement, or query formulation implicitly position the act of starting as trivial or self-evident. The IEP as a conceptual framing identifies this moment as the core challenge. As a result, these paradigms can be said to address usability within exploration, while remaining structurally incapable of initiating exploration under the epistemic conditions of the barriers of the IEP.

Importantly, this is not a claim that such systems are ineffective or poorly designed. Rather, it is a claim about temporal alignment: the problems they are designed to solve occur later in the interaction lifecycle than the IEP itself. This explains why multiple, independently developed systems across the Semantic Web and Digital Humanities communities converge on similar limitations when deployed for lay users encountering a KG for the first time (e.g., SPARKLIS, PG-Explorer, KGViz, Sampo UI [53]).

### 4.4. The Missing Interaction Primitive: Scope Revelation

The analysis above reveals a structural gap in the design space of KG exploration interfaces. While existing paradigms provide strong support for query formulation, schema inspection, and exploratory navigation, they lack an interaction primitive whose primary epistemic function is scope revelation

without user specification. That is, there is no dominant design pattern whose purpose is to communicate what the KG contains and what kinds of questions it can answer before the user is required to act.

Within the IEP framing, such a primitive must satisfy three conditions. First, it must not require the user to articulate a goal or query. Second, it must not rely on the user interpreting ontological structures or terminology. Third, it must function as a meaningful conceptual foothold from which further exploration can proceed. Existing primitives fail to meet these conditions simultaneously, explaining why scope uncertainty persists even in interfaces that successfully mitigate ontology opacity or query incapacity.

This observation reframes the problem of KG exploration at first contact as one of entry-point design. The absence of an appropriate entry-point primitive is not accidental but reflects a broader tendency in interface design to privilege interaction over orientation as surfaced in Section 2. Addressing the IEP therefore requires not incremental improvements to existing paradigms, but the introduction of interaction constructs explicitly designed for epistemic onboarding.

In practice, scope revelation may take multiple forms. These include curated entry questions and answers [54] that expose answerable entities and relations; semantic preview panels that summarise available entity types and relationships; and guided entity anchors that present representative people, places, or events as navigational entry points. Each functions by exposing dataset scope prior to requiring user-formulated queries.

## 4.5. Implications for IEP-Oriented Interface Design

By treating the IEP and its framing as a design constraint rather than a usability flaw, it becomes possible to articulate clearer criteria for what constitutes an IEP-oriented interface. Such interfaces must incorporate interaction primitives whose function is not to respond to user intent, but to help form that intent by revealing the scope and structure of the KG in cognitively accessible terms.

This perspective also clarifies the role of hybrid interface solutions. Interfaces that combine traditional exploration mechanisms with explicit entry-point scaffolding can support both initiation and sustained exploration, provided that these components are temporally and conceptually separated. Ongoing work has built on this insight by introducing the Curated Question and Answer (CuQA) as a concrete instantiation of a scope-revealing interaction primitive that also addresses ontology opacity and query incapacity, and by presenting the Tús Maith framework as a means of supporting their creation and curation at scale [54, 55].

# 5. Examining KG Exploration Interfaces through the Lens of the IEP

In the following subsections, existing KG exploration solutions will be examined through the lens of addressing the barriers characterised in the IEP framing. These barriers are inherently overlapping, and individual solutions often address more than one. However, the core limitation across existing approaches is not the absence of partial solutions, but the absence of approaches that address all barriers simultaneously.

## 5.1. Scope Uncertainty and Ontology Opacity

A large number of interfaces have been developed to support exploration over KGs, but these typically fail to address the fundamental issues of scope uncertainty and ontology opacity that are characterised in the IEP framing. Faceted browsers and search interfaces, such as SampoUI [53], require users to already know enough about the domain to make the first meaningful decision. They do not present a clear or pedagogically informed starting point, and users are instead expected to choose a facet or construct a search term without any understanding of what the KG can or cannot answer. It is not a flaw of faceted browsers that they fail to address the IEP; rather, their design assumes the user has already overcome it. This analysis reveals that they are effective tools for within-exploration support but are structurally incapable of solving the first-contact problem. In other words, within these systems,

the actual scope of the KG is only revealed after a user has already begun interacting with it, which presupposes the very knowledge the user is likely to be missing. As a result, faceted search places the cognitive burden of orientation on the user rather than providing support at the moment it is most needed.

Graph summary and overview visualisation tools, such as ViziQuer [56], attempt to give users a high-level view of the structure of a graph. While these tools can be powerful for technically experienced users, they are often overwhelming for non-technical audiences, particularly in the case of large or semantically dense KGs such as those built in the humanities. Visualisations that attempt to summarise millions of triples inevitably introduce visual and cognitive complexity, and users can struggle to extract meaningful starting points from these views. Moreover, the diagrams themselves are often semantically opaque: they display node and edge types defined in complex ontologies, leaving non-expert users unable to interpret what the structures mean or how they relate to the questions they wish to ask. Thus, while visual summaries aim to provide orientation, they frequently reproduce the problem of ontology opacity rather than alleviating it.

## 5.2. Query Incapacity

Several systems seek to lower the barrier of query formulation, but these solutions focus primarily on the mechanics of translating questions rather than helping users understand what questions are possible to ask in the first place. Natural language to SPARQL interfaces, such as KG-GPT [57] or large language model (LLM) -based pipelines for generating SPARQL over federated KGs [58], best illustrate this. While they remove the burden of writing SPARQL syntax, they still rely entirely on the user knowing what to ask. If a user has no initial sense of the KG's scope (what classes, properties, or event types exist) they cannot form the natural-language questions (NLQs) that these systems expect. In this way, such interfaces address query incapacity but leave scope uncertainty, as characterised in the IEP framing, untouched. As a result, they become useful only after the user has already overcome the hardest part of the exploration process.

Visual query builders, such as KGViz [51], offer users a drag-and-drop interface to construct queries, but they too depend on prior understanding of the underlying schema. A user must know which classes or relationships are appropriate for the query they wish to form, and visual composition does not remove the semantic burden associated with choosing the correct building blocks. This again illustrates ontology opacity: even when the interface hides SPARQL syntax, it cannot hide the conceptual structure of the ontology itself. Consequently, these tools reduce some technical friction but still require users to possess a level of schema awareness that non-technical users are unlikely to have when first approaching a KG.

## 5.3. Starting Point Solutions

Yet other work attempts to provide users with a starting point, but these solutions tend to be either too technical, too reactive, or too narrow in scope to meaningfully address the IEP. Pre-written SPARQL queries, for instance, are often included as part of KG documentation or technical demonstrations [59]. While these queries illustrate what the KG is capable of returning, they are rarely designed with pedagogical intent. They assume that (1) the user is capable of understanding SPARQL and (2) that they can mentally map a technical query onto the domain concepts they care about. They provide little guidance in understanding why a query is meaningful, how it relates to the KG's structure, or what broader set of questions might be possible.

Recommendation systems [38] represent another class of "starting point" solutions, but they suffer from a fundamental mismatch with the IEP. Much like the cold-start problem [60] in recommender systems, a user with no previous engagement provides no basis upon which recommendations can be meaningfully generated. In the KG context, the user is "cold" not because their behavioural history is missing, but because their cognitive orientation is missing: they do not yet know what the KG contains, nor how to ask for information within it. Scope uncertainty is therefore analogous to the

user's cognitive cold start. Any recommendation mechanism that relies on user activity or relevance feedback is inherently reactive and can only become useful once the user has already begun exploring. It cannot address the core challenge of helping users take that first step.

OnSET [44] provides a different approach offering a topic-guided visual query building interface. Although it does not explicitly address the IEP, it implicitly attempts to tackle aspects of the problem as characterised in the IEP framing by reducing ontology opacity: by surfacing topics and visual structures, it helps users build a mental model of the KG's underlying schema. However, OnSET does not directly address scope uncertainty, as characterised in the IEP framing, as users still need to choose which topic to begin with. It also does not fully mitigate query incapacity, as users ultimately must understand how to construct a query using the visual interface. In this sense, OnSET offers a structural entry point into a KG that supports open-ended exploration, leaving users to decide how to navigate the space.

## 6. Limitations

This paper's contribution is conceptual: it defines and frames the IEP rather than empirically validating it here. The barriers have been examined through user studies as part of the corresponding author's recently completed PhD thesis [61], but broader empirical validation across domains and interface types remains necessary to establish prevalence, boundary conditions, and measurable impact.

## 7. Summary and Implications

This paper has introduced and defined the Initial Exploration Problem (IEP), and developed an IEP framing that theorises it as a distinct, critical barrier faced by lay users at the precise moment of first contact with an unfamiliar, complex KG, a problem previously only addressed fragmentarily in both literature and in system design. Through the IEP framing, the problem has been distilled into three interdependent characteristics: *Scope Uncertainty*, *Ontology Opacity*, and *Query Incapacity*. By examining the theoretical basis from which the problem arises, IEP framing positions it within, yet distinct from, established concepts like ASK, exploratory search, and cognitive load theory, as summarised in Table 1. Its uniqueness lies in its temporality (applying strictly at first contact) and its pre-conditional nature (assuming no starting point or explicit goal). The contribution of the IEP framing is therefore not the identification of new individual barriers, but the isolation and characterisation of their convergence at a specific temporal point in interaction: the moment of first contact. While the IEP is most visible in curiosity-driven exploration, it also applies when lay users possess only vague or underspecified goals, insofar as they lack the knowledge required to identify a meaningful entry point into the KG.

Defining the IEP and developing its conceptual framing has two primary implications. Firstly, it establishes a clear conceptual lens and vocabulary through which to evaluate how systems support (or fail to support) the initial exploration phase. It is now possible to assess existing solutions and interface primitives based on how effectively they mitigate Scope Uncertainty, clarify Ontology Opacity, and bypass Query Incapacity. In doing so, it follows a long-standing trajectory in HCI: that of moving from ad hoc design responses to theory-informed interface paradigms, ultimately improving both usability and explanatory power. Secondly, it provides the foundation for the design of targeted scaffolding techniques such as Curated Natural Language Questions and Answers (CuQAs) [54, 55]. By first isolating and naming the problem, the IEP framing supports a shift from ad hoc interface design to theory-informed intervention, aiming to transform the initial moment of confusion into a structured and supported entry into exploration.

## Acknowledgments

The research conducted in this publication was primarily funded by the Irish Research Council under project ID GOIPG/2021/116, partially funded by the School of Computer Science and Statistics Trinity

College Dublin, and supported by the Research Ireland ADAPT Centre for Digital Content Technology (grant number `13/RC/2106_P2`).

## Declaration on Generative AI

The authors used GPT-5 to assist with sentence-level editing to improve grammar, spelling, and clarity. All AI-generated content was manually reviewed, modified as necessary, and validated by the authors, who take full responsibility for the entire manuscript's content and correctness.

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
