# OpenReview forum: "The Initial Exploration Problem in Knowledge Graph Exploration"
_SEMANTiCS.cc/2026/Workshop/UKG — SEMANTiCS 2026 Workshop UKG Submission_

### Official Review · ~Ahmet_Soylu4 · 2026-07-03
**The Initial Exploration Problem in Knowledge Graph Exploration**

**Rating:** 7
**Confidence:** 4

**Review:**

The paper’s contribution is to define the Initial Exploration Problem (IEP) as a distinct first-contact challenge in knowledge graph (KG) exploration. It shows that lay users struggle not only with querying, but with understanding the graph’s scope, structure, and possible entry points. By framing this through scope uncertainty, ontology opacity, and query incapacity, the paper offers a clearer perspective for evaluating KG interfaces and argues for scope revelation as a missing design primitive.

*** Strengths
- The paper identifies a clear and under-specified problem in KG usability, i.e., the difficulty of beginning exploration before the user has a goal or query.
- The three-part framing, i.e., scope uncertainty, ontology opacity, and query incapacity, is useful and helps distinguish the IEP from broader exploratory search or general usability issues.
- The argument that existing KG interfaces often assume a starting point is convincing and gives the paper a clear design implication.
- The proposed idea of scope revelation is a strong contribution because it points toward concrete future interface design and evaluation criteria.

*** Weaknesses
- The paper is mainly conceptual, so the IEP would be stronger with more direct empirical evidence or user examples showing the problem in practice.
- Some parts are repetitive, especially in explaining the three barriers and how existing systems fail to address them.
- The distinction between IEP and related concepts such as onboarding, exploratory search, and ASK is useful but could be sharpened further.
- The design contribution could be more concrete: “scope revelation” is promising, but the paper could explain more clearly how it should be implemented or evaluated.

---

### Official Review · ~Tom_De_Nies1 · 2026-07-22
**Interesting problem, but lacking depth**

**Rating:** 6
**Confidence:** 4

**Review:**

This paper presents a conceptual definition of the Initial Exploration Problem (IEP), applied to Knowledge Graphs, and a limited review of existing approaches that address part(s) of this problem.

While the authors claim to make four contributions, I would argue that that is a stretch. The contributions follow each other logically, combining into one actual contribution in my opinion. Since the authors have published several papers before on the same topic, this seems like a compartmentalization of the theoretical side of their research, whereas the other papers focus on specific implementations to address the IEP. The paper reads as an introductory chapter of a PhD thesis, which seems to be confirmed in Section 6. While well-written, I do think it would be more valuable when combined with the actual applications and findings, rather than a standalone paper.

The purpose of the sections up to 3.4 seems to be primarily to provide context and motivation for the research question, but could benefit from more brevity in my opinion. From 3.4 onwards, we get to the "meat" of the paper, and the authors do provide a useful definition of the IEP, stressing that it is a three-pronged problem, whereas most existing research focuses on a singular aspect.

In section 4, I'm not sure I agree with the statement "The IEP is temporally-bounded: it presents itself at the very start of KG exploration."
I might be splitting hairs, but this statement assumes that the knowledge graph is limited to one domain, one use case, one distinct graph, one ontology, etc. I would argue that the IEP presents itself at multiple occasions, until the entire scope of the KG has been explored. Also, is it relevant that the problem is (or is not) temporally bounded? Later in the text, it seems that this is primarily used to indicate why existing approaches to address KG exploration don't solve the IEP. One could make the same claim by simply stating that it is epistemically bound.

In section 4.4, it is claimed that "there is no dominant design pattern whose purpose is to communicate what the KG contains and what kinds of questions it can answer before the user is required to act". This is untrue. Proper knowledge graph design recommends specification of data shapes and constraints for validation, using technologies such as OWL and/or SHACL. I'm not saying it is necessarily suitable for human consumption, but the pattern does exist. It is only when the knowledge graph lacks these specifications that such a communication is difficult to formulate.

After spending all of section 4 on which exploration approaches do not entirely address the IEP, it is stated in 4.5 that we can use the IEP to "articulate clearer criteria for what constitutes an IEP-oriented interface". I would have liked to actually see measurable details of these criteria listed, as that would be a really meaningful contribution, and would serve as a proper evaluation framework for the approaches discussed in section 5. When are the criteria sufficiently satisfied? Additionally, I was disappointed to see that each discussed approach in section 5 is again categorized under one of these criteria. Is there truly no existing approach that combines any of these?

The paper leaves the reader with a lot of open questions, and I must admit that after reading it, I'm not really closer to a suitable design for an interface that mitigates the IEP.

In summary: I think the paper addresses an important issue, and offers a clear situation and definition of the IEP, illustrated by the shortcomings of the existing approaches that were mentioned. However, I am in doubt whether the contribution merits a 15-page paper, or would be more suitable as a position paper. It will certainly be an interesting topic to discuss during the workshop, so I am inclined to accept it as a borderline paper, but would also not oppose a reject.